# Effects of additional oral theophylline with inhaled therapy in patients with stable chronic obstructive pulmonary disease: A systematic review and meta-analysis

Qiang Yang ⓘ*, Pingxiu Tang, Xunyan Zhang

Department of Pharmacy, Suining Central Hospital, Suining City, Sichuan Province, China

* 1247910520@qq.com

## Abstract

The rationale for additional treatment of oral theophylline with inhaled therapy in patients with stable chronic obstructive pulmonary disease (COPD) is unclear. The databases including The Cochrane Library、PubMed、Embase and Web of Science were searched to collect randomized controlled trials (RCTs) involving the inhaled therapy plus additional theophylline therapy for the treatment of patients with stable COPD up to December 31, 2023. The forced expiratory volume in 1 second ($FEV_1$)、forced expiratory volume in 1s% predicted ($FEV_1$% pred)、forced vital capacity (FVC)、$FEV_1$/FVC%、peak expiratory flow rate(PEFR)、exacerbation rate of COPD、COPD related hospital admissions、total symptom score and drug-related adverse reactions were extracted from literatures and the meta-analysis was conducted using the RevMan 5.4 software. 10 RCTs involving 2771 patients were included. The meta-analysis results showed that additional theophylline improved $FEV_1$ with MD 0.08 (95% CI: 0.06 to 0.09, p<0.00001)、FVC with MD 0.13 (95% CI: 0.10 to 0.15, p<0.00001), reduce the risk of exacerbation rate with OR 0.75 (95% CI: 0.60 to 0.94, p=0.01) and COPD related hospital admissions with MD -0.07 (95% CI: -0.13 to -0.01, p=0.01). However, there was no significant difference in $FEV_1$% pred with MD 0.45 (95% CI: -1.41 to 2.30, p=0.64)、$FEV_1$/FVC% with MD -0.24 (95% CI: -3.26 to 2.79, p=0.88) and total symptom score with MD -0.03 (95% CI: -0.14 to 0.09, p=0.65). Furthermore, additional theophylline therapy induced a high incidence of drug-related adverse reactions with OR 1.33 (95% CI: 1.12 to 1.58, p=0.001), especially in gastrointestinal adverse reactions. Oral theophylline could be a supplementary therapeutic option when inhaled therapy is insufficient regarding of improvement in pulmonary function and reducing in exacerbation risk. However, additional theophylline therapy could increase the risk of drug-related adverse reactions and should be concerned.

COPD is a progressive complex lung disease and cause a significant burden on the healthcare system worldwide [1–2]. Inhaled bronchodilators, which could stabilize disease、improve symptoms and reduce the frequency of exacerbations [3–4], play important roles in treatment

**Data availability statement:** All relevant data are within the manuscript

**Funding:** The author(s) received no specific funding for this work.

**Competing interests:** The authors have declared that no competing interests exist.

of stable COPD. Theophylline, belonging to methylxanthine, has been a conventional oral bronchodilator for more than 100 years in COPD treatment [5]. Previous clinical studies indicated that theophylline could improve respiratory function and alleviate respiratory symptoms by strengthening of respiratory muscles, enhancing response to hypoxic ventilation, and thus increasing tidal volume [6–10]. Nowadays, guidelines suggest that theophylline has limitations in terms of its less efficacy in comparison with novel inhaled bronchodilators、narrow therapeutic window and adverse reactions [3]. The current evidences regard theophylline as third-line treatment of COPD and may be useful if the other choices are unavailable or unaffordable [11]. However, the oral theophylline now is commonly used as an additional agent to inhaled therapy in treatment of stable COPD, which bring about uncertain benefits but more risks of adverse reactions. Previous meta-analysis has shown that compared to placebo, theophylline combined with inhaled $\beta_2$-receptor agonists could improve $FEV_1$ and respiratory symptoms [12]. However, there is no systematic evaluation in additional therapy effect of theophylline on inhaled therapy. Furthermore, theophylline has relatively narrow therapeutic range and may frequently induce adverse reactions such as gastrointestinal、cardiovascular and neurological adverse reactions. However, it is unclear whether additional theophylline on inhaled therapy could increase adverse reactions. Therefore, we conducted a meta-analysis to systematically evaluates the effects of additional theophylline with inhaled therapy for stable COPD patients, aiming to provide reference for clinical treatment.

## 1 Methods

### 1.1 Eligibility criteria and search

The meta-analysis was performed followed the guidance of the Preferred Reporting Items for Systematic Reviews and Meta-Analyses (PRISMA) [13].

We searched for RCTs in the PubMed、Embase、Web of Science、The Cochrane Library from the inception of the database to the present date (December 31, 2023) to retrieve published studies involving the additional theophylline treatment combined with regular inhaled therapy in stable COPD. We also manually searched for references to ensure that more literatures were included. The relevant retrieval strategy was as follows: ("Pulmonary Disease, Chronic Obstructive" OR "Chronic Obstructive Lung Disease" OR "COPD" OR "Chronic Obstructive Pulmonary Diseases") AND ("Theophylline" OR "1,3-Dimethylxanthine" OR "Armophylline"OR "Elixophyllin" OR "Uniphyllin") AND ("randomized controlled trial" OR "random*" OR "placebo").

### 1.2 Inclusion criteria

(1) stable COPD; (2) age>18 years old; (3) RCTs; (4) no limitations in gender, age, ethnicity, or treatment duration; (5) without bronchiectasis、tuberculosis、cancer、heart failure and infections.

### 1.3 Exclusion criteria

(1) non-stable COPD; (2) non-RCTs; (3) Valid ending data unable to be extracted; (4) Full text of the study is not available;(5) with bronchiectasis、tuberculosis、cancer、heart failure and infections.

### 1.4 Data extraction

The characteristics of literature including first author、year of publication、country、participant gender、intervention、duration and main outcomes were collected. The main

outcomes include:(1) $FEV_1$; (2) $FEV_1$%pred; (3) FVC; (4) $FEV_1$/FVC%; (5) PEFR; (6) exacerbation rate of COPD patients; (7) COPD related hospital admissions; (8) total symptom score; (9) drug-related adverse reactions. The outcomes selected to analysis were of importance to evaluate the therapeutic effects. COPD exacerbation rate was defined that the proportion of COPD patients with acute exacerbation to the total number COPD patients. The exacerbation included emergency visits and hospital admissions.

### 1.5 Data collection process and risk of bias assessment

Two researchers screened the included literature independently based on inclusion and exclusion criteria. For articles with difference between two researchers were judged by other researcher to identify inclusion or exclusion. And then the data were extracted from the literatures by two authors. At the same time, the bias risk of RCTs were evaluated by two researchers according to the Cochrane manual [14].

### 1.6 Synthesis of results

Analysis was in accordance with the intention-to-treat principle. The meta-analyses were conducted by RevMan 5.4 software. The odds ratio (OR) with 95% CI of dichotomous data were computed by Mantel-Haenszel method, and the mean difference (MD) with 95% confidence intervals (CI) of continuous data were computed by inverse variance method. Heterogeneity testing was conducted through Q-test. If $P \geq 0.1$ and $I^2 \leq 50\%$, indicating no heterogeneity, meta-analyses were performed with fixed effects model; If $P < 0.1$ or $I^2 > 50\%$, indicating existence of heterogeneity, meta-analyses were performed by random effects model.

## 2 Results

### 2.1 Study selection and characteristics

As shown in Fig 1, we have retrieved 935 articles with PubMed (n=252)、Embase (n=360)、The Cochrane library (n=204) and Web of Science (n=119). 341 duplicates were excluded through Endnote software and then 574 articles were excluded by reading the title and abstract. Next, we assessed 20 full-text articles for eligibility and excluded 10 articles. Finally, we included 10 RCTs in the meta-analysis [15–24]. The characteristics of the included studies are presented in Table 1.

### 2.2 Risk of bias

The risk of bias of the included studies is presented in Fig 2. All the RCTs had an unclear risk bias for allocation concealment (selection bias). All studies had patient dropouts during the follow-up period and the analysis was in accordance with the intention-to-treat or Per-Protocol principle. We judged that there was no study with a high risk of bias.

### 2.3 Outcome of Meta-analysis

Four studies [16,17,19,22] with 1825 patients reported $FEV_1$% pred, which showed no significant difference in the change of $FEV_1$% pred with MD 0.45 (95% CI: -1.41 to 2.30, p=0.64, $I^2 = 0\%$) (Fig 3A). Seven studies [17,18,20–24] with 1086 patients reported $FEV_1$ showing a significant improvement of $FEV_1$ in theophylline group with MD 0.08 (95% CI: 0.06 to 0.09, p<0.00001, $I^2 = 50\%$) (Fig 3B). Five studies [17,18,20,23,24] with 996 patients reported FVC with a result of significant improvement of FVC in theophylline group with MD 0.13 (95% CI: 0.10 to 0.15, p<0.00001, $I^2 = 56\%$) (Fig 3C). Three studies [17,22,23] with 234 patients reported $FEV_1$/FVC%, which showed no significant difference in the change of $FEV_1$/FVC%

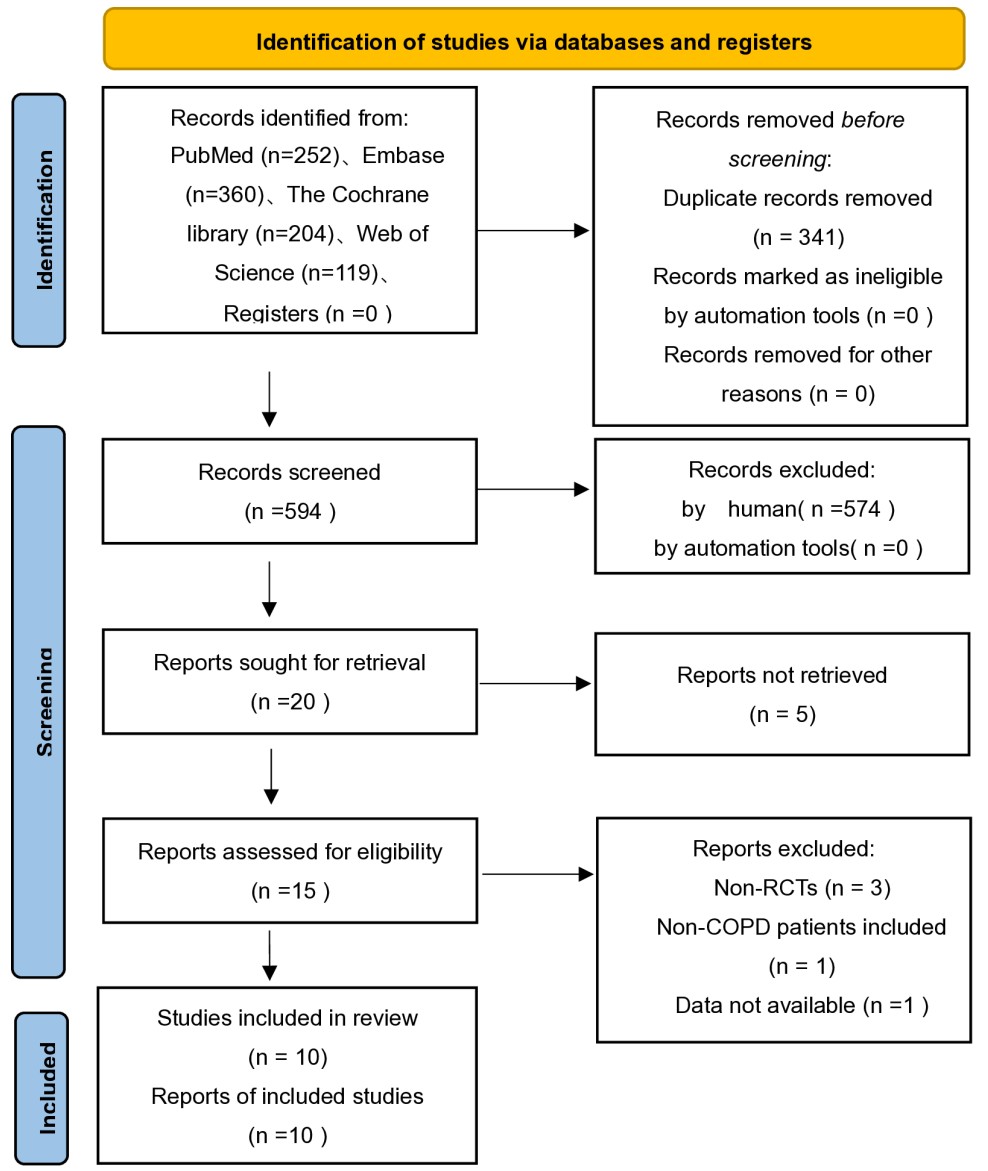

**Fig 1. Flow diagram of study selection.**

in the two groups with MD -0.24 (95% CI: -3.26 to 2.79, p=0.88, $I^2$ = 0%) (Fig 3D). Two studies [18–19] with 203 patients reported PEFR, presenting a significant improvement of PEFR in theophylline group with MD 11.69 (95% CI: 10.04 to 13.35, p<0.00001, $I^2$ = 0%) (Fig 3E). Two studies [15–16] with 1637 patients reported COPD related hospital admissions, showing that inhalation therapy combined with theophylline could reduce the COPD related hospital admissions with MD -0.07 (95% CI: -0.13 to -0.01, p=0.01, $I^2$ = 0%) (Fig 3F). Four studies [15–17. 24] with 2450 patients reported exacerbations rate of COPD patients, indicating that inhalation therapy combined with theophylline could significantly reduce the exacerbations rate with OR 0.75 (95% CI: 0.60 to 0.94, p=0.01, $I^2$ = 0%) (Fig 3G). Three studies[21. 23–24] with 687 patients reported total symptom score, which showed no significant difference in the change of total symptom score in the two groups with MD -0.03 (95% CI: -0.14 to 0.09,

**Table 1. Basic characteristics of included articles.**

| Study | Country | N | | Male/Female | | Interventions | | Theophylline Concentrations | Duration | Outcomes |
|---|---|---|---|---|---|---|---|---|---|---|
| | | T | C | T | C | T | C | | | |
| Borja G. Cosío 2016[15] | Spain | 36 | 34 | 30/6 | 27/7 | Inhaled fluticasone propionate 500 µg and salmeterol 50 µg twice daily + oral theophylline | Inhaled fluticasone propionate 500 µg and salmeterol 50 µg twice daily +placebo | 3.96 ± 3.5µg/ml | 52 weeks | ⑥⑦ |
| Graham Devereux 2018[16] | Britain | 788 | 779 | 425/363 | 418/361 | Inhaled therapy (including inhaled corticosteroids(ICS)/LABA、ICS/LAMA、ICS/LABA/LAMA) + oral theophylline | Inhaled therapy (including ICS/LABA、ICS/LAMA、ICS/LABA/LAMA) | 1~5µg/ml | 52 weeks | ①⑥⑦⑨ |
| Xiao-feng Xiong 2018[17] | China | 85 | 85 | 70/15 | 68/17 | Tiotropium 18 µg once daily + oral theophylline 100 mg twice daily | Tiotropium 18 µg once daily | – | 6 months | ①②③④⑦⑨ |
| V.B ELLIA 2002[18] | Italy | 75 | 80 | 64/11 | 71/9 | Inhaled oxitropium bromide 200mg bid + oral theophylline 300mg bid | Inhaled oxitropium bromide 200mg bid+ theophylline | 8~20µg/ml | 8 weeks | ②③⑤⑨ |
| Koichi Nishimura 1995[19] | Japan | 24 | 24 | 24/0 | 24/0 | Inhaled salbutamol 400µg + ipratropium bromide 80µg qid + oral theophylline | Inhaled salbutamol 400µg + ipratropium bromide 80µg qid. | 15±5.5 µg/ml | 4weeks | ①⑤⑧ |
| Koichi Nishimura 1993[20] | Japan | 12 | 12 | 11/1 | 11/1 | Inhaled salbutamol 200µg + ipratropium bromide 40µg qid + oral theophylline | Inhaled salbutamol 200µg + ipratropium bromide 40µg qid | 7.7±2.2 µg/ml | 4weeks | ②③⑤⑧ |
| Subramanian 2015[21] | India | 24 | 26 | 21/3 | 25/1 | Inhaled formoterol 24µg plus budesonide 800µg daily + oral theophylline | Inhaled formoterol 24µg plus budesonide 800µg daily | – | 60days | ②⑧ |
| CJ.CLARK 1980[22] | Britain | 20 | 20 | 11/9 | 11/9 | Oral aminophylline 500mg bid +salbutamol 4 mg three times daily | salbutamol 4 mg three times daily | – | 3weeks | ①②④ |
| Peter Thomas 1992[23] | Canada | 12 | 12 | 6/6 | 6/6 | Inhaled salbutamol 200 µg tid + oral aminophylline | Inhaled salbutamol 200 µg tid | 10-16.5 | 14days | ②③④⑧ |
| Richard L 2001[24] | America | 313 | 310 | 210/103 | 198/112 | Inhaled salmeterol 42 µg bid + oral theophylline twice daily | Inhaled salmeterol 42 µg bid | 10 - 20µg/mL | 12 weeks | ②③⑥⑦⑧⑨ |

Abbreviations: T, test group; C, control group; -, not mentioned; ①FEV$_1$% pred; ②FEV$_1$; ③FVC; ④FEV$_1$/FVC%; ⑤PEFR; ⑥exacerbations rate of COPD patients; ⑦COPD related hospital admissions; ⑧total symptom score; ⑨drug-related adverse reactions.

p=0.65, I$^2$ = 95%) (Fig 3H). Four studies [16–18,24] with 2535 patients reported drug-related adverse reactions, which showed that inhalation therapy combined with theophylline could significantly increase drug-related adverse reactions with OR 1.33 (95% CI: 1.12 to 1.58, p=0.001, I$^2$ = 39%) (Fig 3I). From the cumulative occurrence frequency, the main adverse reactions were gastrointestinal adverse reactions、cardiovascular adverse reactions and neurological adverse reactions, as shown in Table 2. Four studies [16–18,24] with 1882 patients reported gastrointestinal adverse reactions, which showed that inhalation therapy combined with theophylline could significantly increase gastrointestinal adverse reactions with MD 1.37 (95% CI: 1.12 to 1.66, p=0.002, I$^2$ = 43%) (Fig 3J). However, there was no significant difference between the two groups in terms of cardiovascular [16–18] and neurological [16–18,24] adverse reactions with MD 0.96 (95% CI: 0.78 to 1.17, p=0.68, I$^2$ = 0%) (Fig 3K) and MD 0.76 (95% CI: 0.51 to 1.15, I$^2$ = 0%) (Fig 3L), respectively.

## 3 Discussion

In the present systematic review and meta-analysis, the additional effect of theophylline on a background of regular inhalation treatment in stable COPD was evaluated with pulmonary function、symptom、exacerbation and drug-related adverse reactions. As respect of

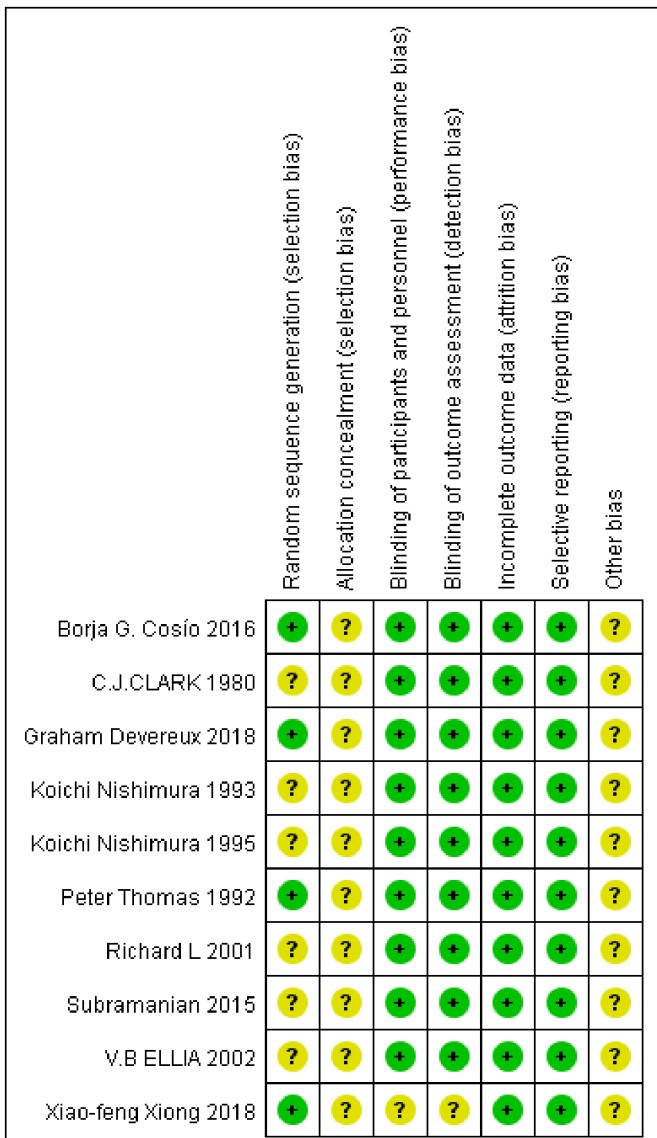

**Fig 2. Summary of the risk of bias.**

pulmonary function, the meta-analysis results showed a significant improvement in $FEV_1$、 FVC and PEFR but no difference in $FEV_1$% pred and $FEV_1$/FVC%. Furthermore, the results showed that additional theophylline therapy could reduce the exacerbation rate and COPD related hospital admissions. However, there was no difference in total symptom score, indicating that additional theophylline therapy could not improve symptom of COPD patients. In addition, additional theophylline therapy induced a high incidence of drug-related adverse reactions, especially the gastrointestinal adverse reactions. Previous studies about the effect of theophylline on the lung function of stable COPD patients are inconsistent. K Nishimura et al [20] have investigated the additive effect of oral theophylline on combined inhaled both salbutamol and ipratropium bromide therapy and the result indicated that additional oral theophylline could significantly improve the $FEV_1$ and FVC. Xiao-feng Xiong et al [17] have explored the effects of theophylline combined with inhaled tiotropium on small

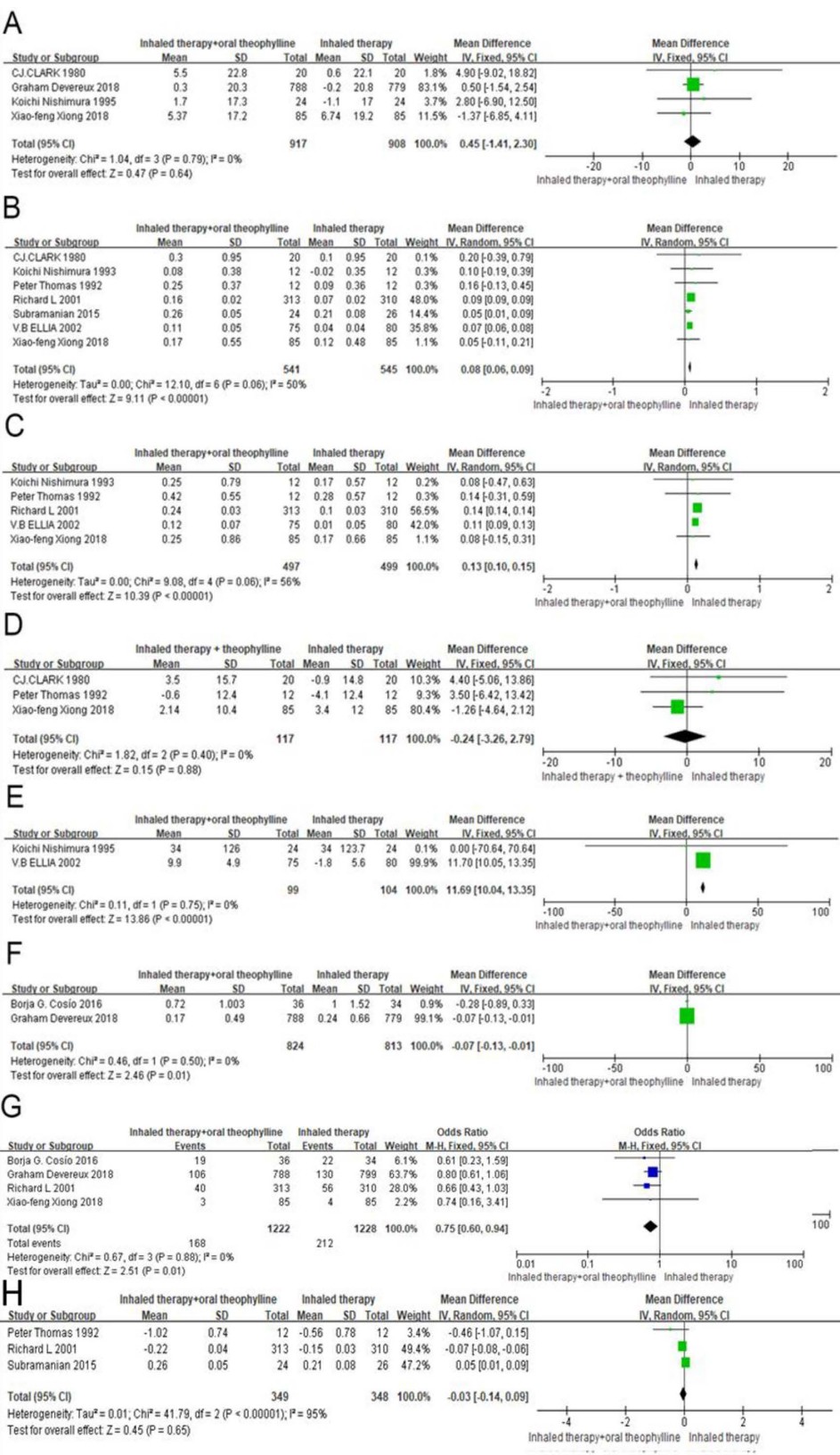

**Fig 3. Results of Meta-analysis: (A)FEV1% pred; (B) FEV1; (C) FVC; (D) FEV1/FVC%; (E) PEFR; (F)hospital admissions; (G)exacerbation rate; (H) total symptom score; (I) adverse reactions (J)gastrointestinal adverse reactions; (K)cardiovascular adverse reactions; (L)neurological adverse reactions.**

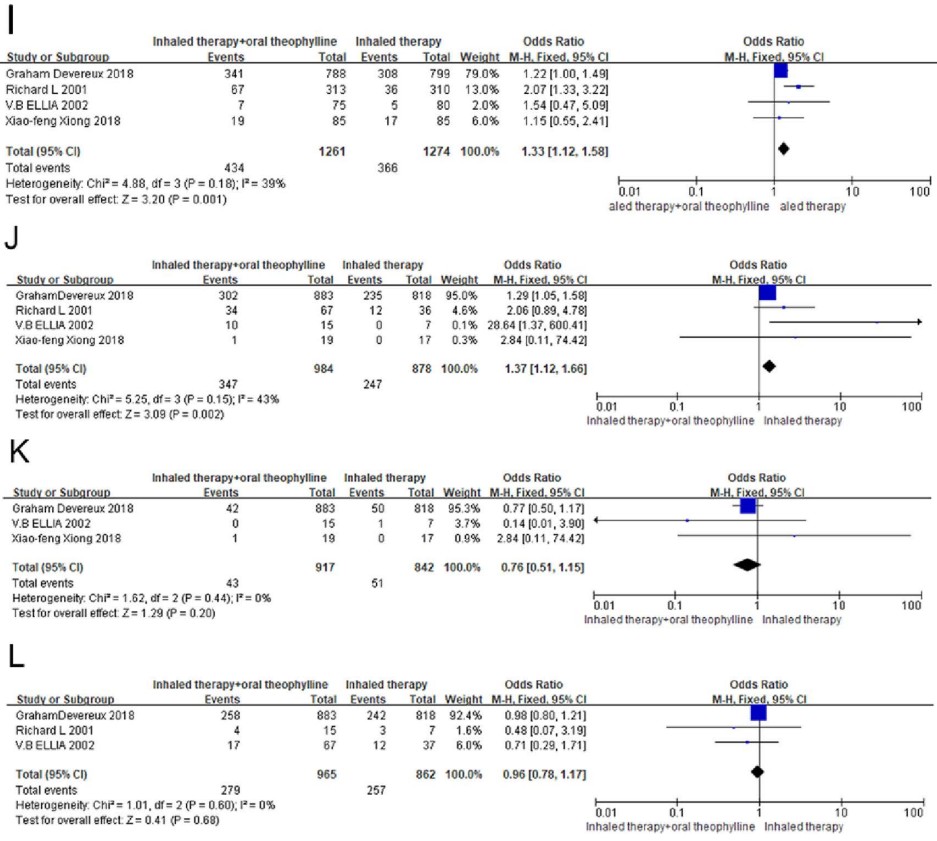

**Fig 3.** Continued.

**Table 2. Summary of adverse reactions.**

| Adverse reaction classification | adverse reaction manifestations |
|---|---|
| Gastrointestinal adverse reactions | nausea、vomiting、reflux、 diarrhea、 abdominal pain、stomach irritation |
| Neurological adverse reactions | insomnia、anxiety、headache、dizziness、agitation、convulsions |
| Cardiovascula adverse reactions | palpitations、arrhythmia、tachycardia |

airway function, finding that additional theophylline could not improve FVC and $FEV_1$. The authors considered that $FEV_1$ decline is related to high airway mucus secretion and inhaled tiotropium could improve $FEV_1$ by antagonizing the cholinergic receptor and then reducing airway mucus secretion but the theophylline has no such effect [25]. Our meta-analysis indicated that the absolute values of $FEV_1$ and FVC were enhanced with additional theophylline therapy, which might be explained in terms of relaxation of smooth muscle and anti-inflammatory effect of theophylline [26]. However, there was no significant change in $FEV_1$% pred and $FEV_1$/FVC%, which are more critical in reflecting the small airway function and are likely to be considered important in clinical practice. The improved absolute value of $FEV_1$ resulted no increase in $FEV_1$% pred and the improvement of both $FEV_1$ and FVC may result no change of $FEV_1$/FVC%. This result was consistent with the total symptom score change. Our meta-analysis revealed that the add-on theophylline had no effect on improving the symptom of COPD patients with no significant difference in total symptom score change, as previous study has demonstrating that the small bronchodilation of theophylline may be insufficient to ameliorate a patient's symptoms [27–28]. However, there was heterogeneity in

symptom scores. Sensitivity analysis using Stata14 software revealed that after excluding the Subramanian2015 study, the heterogeneity was significantly reduced, yet there remained no significant difference in symptom scores between the two groups (MD = -0.14, 95% CI: -0.44 to 0.15, p = 0.35, I²= 37%) (As shown in the Fig 4). The study of Subramanian2015 specifically reported night symptom scores, which may explain the heterogeneity compared to the other two studies. Our meta-analysis indicated non-significant results for FEV1% pred、 FEV1/ FVC% and symptom scores. FEV1% pred and FEV1/FVC% were important index to evaluate lung function and the effectiveness of medication therapy. Theophylline is a weak acting bronchodilator and adding theophylline on the basis of potent inhaled bronchodilators may do not produce additional effects, inducing no improvement in FEV1% pred and FEV1/ FVC%. Symptoms scores are closely related to lung function and non-significant FEV1% pred and FEV1/FVC% improvements result non-significant symptom scores. Exacerbation of COPD could accelerate the deterioration of diseases and reduce the quality of life. Further-more, exacerbation of COPD related hospital admissions could increase economic burden of patients. A randomized, double-blind, double-dummy trial in 943 patients with COPD

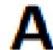

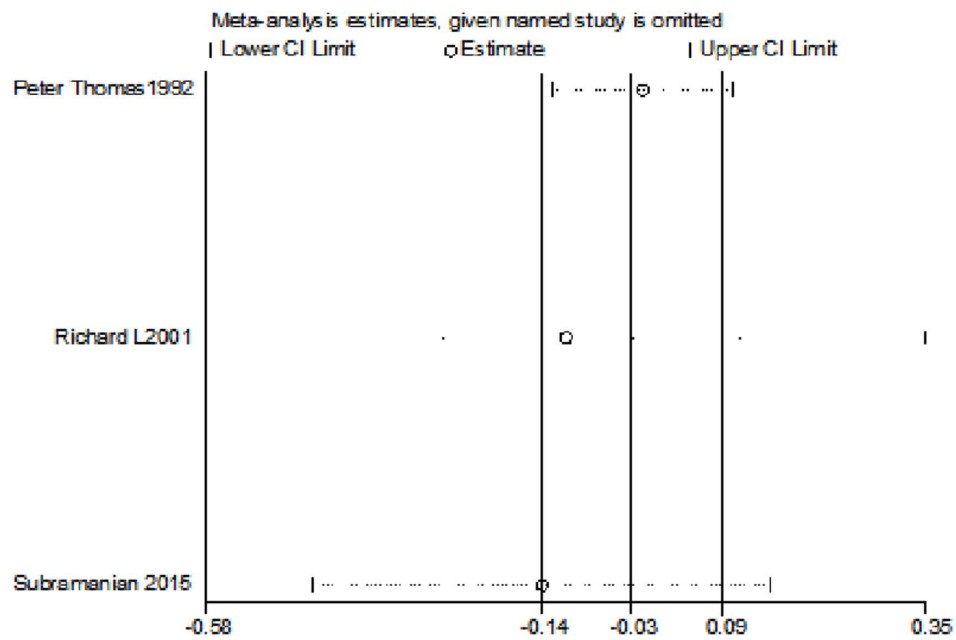

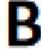

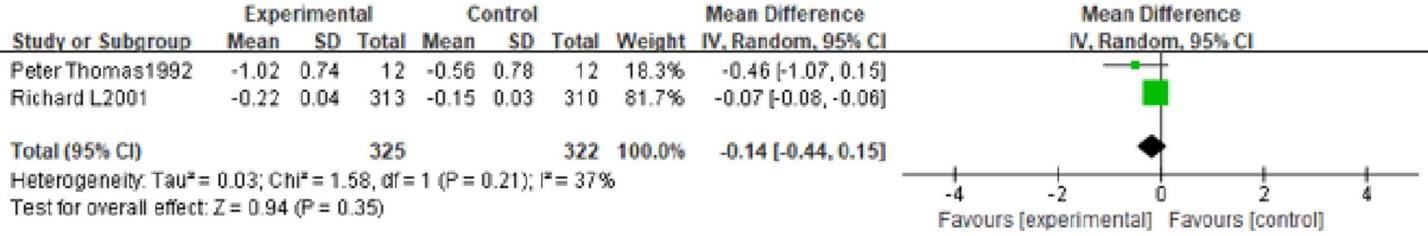

**Fig 4. (A) Sensitivity analysis of total symptom scores; (B) Meta-analysis of total symptom scores after deleting the study of Subramanian2015.**

exploring the effects of combined treatment of salmeterol and theophylline showed that combination treatment resulted in significantly fewer COPD exacerbation compared with theophylline alone, but not with the salmeterol alone [24]. Dennis E Niewoehner et al [29] performed a trail to investigate the risk indexes for exacerbation of COPD patients who inhaled tiotropium during a 6-month follow-up period, finding that theophylline use was an independent risk factors for exacerbations. Johannes Fexer et al conducted a retrospective cohort to investigated the connection between theophylline and the risk of exacerbation, concluding that theophylline treatment elevates the incidence of exacerbation [30]. ICS is widely used in COPD treatment due to its strong anti-inflammatory effect. Previous study showed that theophylline enhanced histone deacetylase activity and potentiated the anti-inflammatory properties of glucocorticoids [31–32], and the effect was also demonstrated by preclinical studies [5]. However, a propensity scores matching analysis indicated that theophylline significantly increased the risk of overall exacerbation [33] when combined with ICS and LABA in treatment of COPD patients. Other RCT also showed that theophylline did not enhance the anti-inflammatory properties of inhaled Fluticasone-Salmeterol and reduce the exacerbation rate of COPD patients during follow-up [19]. The possible causes may be that theophylline enhance immunosuppressive effects of ICS by restoring corticosteroid sensitivity and increase the risk of pneumonia [34–36]. Our meta-analysis result of RCTs showed that additional theophylline therapy to both inhaled bronchodilators and inhaled bronchodilators plus ICS could reduce the exacerbation rate and COPD related hospital admissions, although there was no significant difference in each single study. This result may be explained by that a larger patient size was included in meta-analysis, resulting a significant difference. The funnel plot of exacerbation rate is symmetrical, indicating no publication bias (Fig 5). Therefore, the theophylline may be more suitable for patients who have a high risk of acute exacerbation especially when have already been treated with ICS+LABA+LAMA. Meanwhile, our result also indicated that the additional theophylline therapy induced a high incidence of

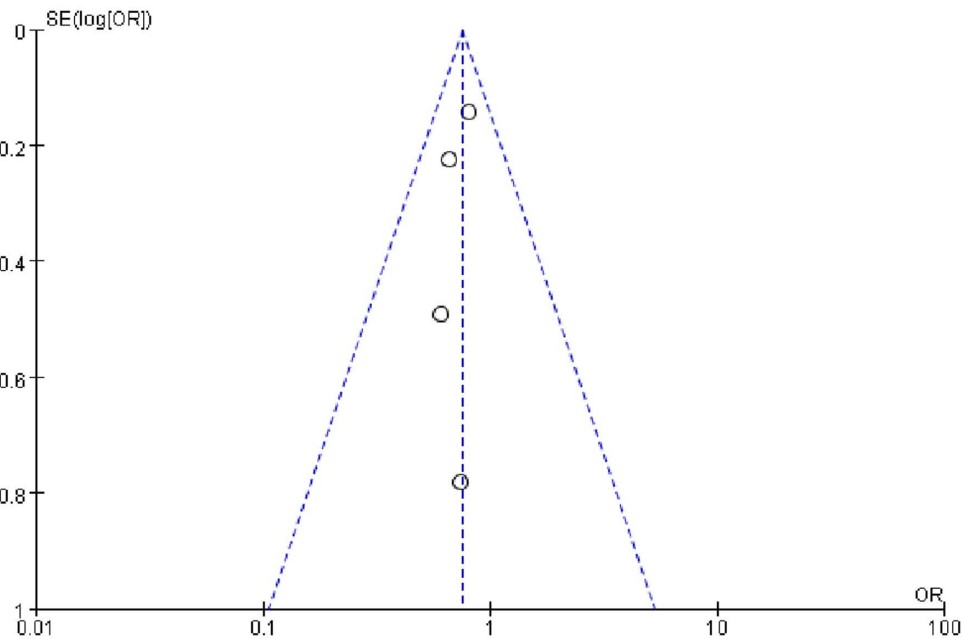

**Fig 5. Funnel plot of exacerbation rate.**

drug-related adverse reactions, which was consistent with other studies [37–38]. The therapeutic window of theophylline is relatively narrow (5~20 μg/ml) and show high rate of adverse reactions even though within the therapeutic window (15~20 μg/ml) such as nausea、vomiting、irritability and insomnia. When the serum concentration exceeds 20μg/ml, tachycardia and arrhythmia may occur. Fever、dehydration、convulsions and even death caused by respiratory and cardiac arrest may occur when serum concentration exceeds 40μg/ml. In addition, many factors could affect the serum concentration of theophylline, such as age、 -medication and disease. Theophylline is metabolized by cytochrome P450 mixed function oxidases. Many other drugs can modify theophylline. For example, macrolides and quinolones could decline the clearance of theophylline and may lead to an increase of serum concentration, rising the risk of adverse reactions of theophylline. Therefore, clinician need to be in cautions when prescribing theophylline and it is recommended to monitor the serum concentration. There is still limitation in our study. For example,Meaningful subgroup analyses were not conducted due to significant challenges in classifying the literature with regard to inconsistencies in study design, population characteristics, and medication regimens. Furthermore, there was no allocation concealment schemes reported in all RCTs and most of RCTs have not report the randomization schemes. Allocation concealment is an important step in randomized controlled trials, aimed at ensuring the confidentiality of the randomization process and preventing researchers from predicting the grouping of subjects before allocation, thereby reducing selection bias. If no allocation concealment scheme, researchers may make subjective judgments during randomization, leading to certain subjects being preferentially assigned to specific groups and resulting in selection bias. The results of meta-analysis depend on the quality of the included studies and the reliability of the randomization process. The lack of allocation concealment schemes in the original research would affect the objectivity and accuracy of the meta-analysis, such as leading to overestimation of theophylline's benefits. Therefore, more high-quality and well-designed RCTs are still needed to evaluate the effects of additional oral theophylline in stable COPD.

## 4  Conclusion

In this manuscript we firstly presented a systematical review and meta-analysis to evaluate the effects of additional oral theophylline to inhaled therapy of stable COPD patients. The results indicated that additional theophylline could improve pulmonary function in $FEV_1$, FVC and PEFR but not in $FEV_1$% pred and $FEV_1$/FVC%, reduce exacerbations rate and COPD related hospital admissions. However, there was a trend of increased drug-related adverse reactions especially in gastrointestinal adverse reactions. Overall, additional oral theophylline exhibited some positive effects in stable COPD patients and could be a supplementary therapeutic option when inhaled therapy is insufficient regarding of improvement in pulmonary function and reducing in exacerbations risk and the drug-related adverse reactions should be concerned. However, the duration of the studies included in the meta-analysis was inconsistent, and the types of COPD were not distinguished. Therefore, researches for long-term assessing cardiovascular risks or theophylline's role in different COPD phenotypes were needed.

## Supporting information

**S1 Data.  Data extracted from the primary research**
(XLSX)

**S2 Data.  List of all 594 sources screened**
(XLSX)

**S3 Data. Numbered table of all studies**
(XLSX)

**S4 Data. Table of the risk of bias**
(XLSX)

**S1 File. Explanation of how missing data were handled**
(DOCX)

**S2 File. PRISMA Checklist**
(DOC)

## Author contributions

**Data curation:** Qiang Yang, Pingxiu Tang, Xunyan Zhang.

**Software:** Qiang Yang.

**Supervision:** Pingxiu Tang, Xunyan Zhang.

**Writing – original draft:** Qiang Yang.

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
