## [Decision Letter · Decision Letter 0]

12 Nov 2024

PONE-D-24-33914Effects of additional oral theophylline with inhaled therapy in patients with stable chronic obstructive pulmonary disease: A systematic review and meta-analysisPLOS ONE

Dear Dr. Yang,

Thank you for submitting your manuscript to PLOS ONE. After careful consideration, we feel that it has merit but does not fully meet PLOS ONE’s publication criteria as it currently stands. Therefore, we invite you to submit a revised version of the manuscript that addresses the points raised during the review process.

We look forward to receiving your revised manuscript.

Kind regards,

Shailza Singh, Ph.D

Academic Editor

PLOS ONE

Journal Requirements: When submitting your revision, we need you to address these additional requirements. 1. Please ensure that your manuscript meets PLOS ONE's style requirements, including those for file naming. The PLOS ONE style templates can be found at https://journals.plos.org/plosone/s/file?id=wjVg/PLOSOne_formatting_sample_main_body.pdf and https://journals.plos.org/plosone/s/file?id=ba62/PLOSOne_formatting_sample_title_authors_affiliations.pdf 2. As required by our policy on Data Availability, please ensure your manuscript or supplementary information includes the following:  A numbered table of all studies identified in the literature search, including those that were excluded from the analyses.   For every excluded study, the table should list the reason(s) for exclusion.   If any of the included studies are unpublished, include a link (URL) to the primary source or detailed information about how the content can be accessed.  A table of all data extracted from the primary research sources for the systematic review and/or meta-analysis. The table must include the following information for each study:  Name of data extractors and date of data extraction  Confirmation that the study was eligible to be included in the review.   All data extracted from each study for the reported systematic review and/or meta-analysis that would be needed to replicate your analyses.  If data or supporting information were obtained from another source (e.g. correspondence with the author of the original research article), please provide the source of data and dates on which the data/information were obtained by your research group.  If applicable for your analysis, a table showing the completed risk of bias and quality/certainty assessments for each study or outcome.  Please ensure this is provided for each domain or parameter assessed. For example, if you used the Cochrane risk-of-bias tool for randomized trials, provide answers to each of the signalling questions for each study. If you used GRADE to assess certainty of evidence, provide judgements about each of the quality of evidence factor. This should be provided for each outcome.   An explanation of how missing data were handled.  This information can be included in the main text, supplementary information, or relevant data repository. Please note that providing these underlying data is a requirement for publication in this journal, and if these data are not provided your manuscript might be rejected. 3. Please include captions for your Supporting Information files at the end of your manuscript, and update any in-text citations to match accordingly. Please see our Supporting Information guidelines for more information: http://journals.plos.org/plosone/s/supporting-information.

**Additional Editor Comments:**

Please revise as per both reviewers suggestions

Reviewers' comments:

Reviewer's Responses to Questions

**Comments to the Author**

1. Is the manuscript technically sound, and do the data support the conclusions?

Reviewer #1: Yes

Reviewer #2: Partly

2. Has the statistical analysis been performed appropriately and rigorously?

Reviewer #1: Yes

Reviewer #2: Yes

3. Have the authors made all data underlying the findings in their manuscript fully available?

Reviewer #1: Yes

Reviewer #2: Yes

4. Is the manuscript presented in an intelligible fashion and written in standard English?

Reviewer #1: Yes

Reviewer #2: Yes

5. Review Comments to the Author

Reviewer #1: The manuscript by Yang et al presented a systematic review and meta-analysis on the effects of additional oral theophylline in conjunction with inhaled therapy in stable COPD patients. The topic is an interesting one in clinical relevance as theophylline is applied in COPD for very long time. The study is well-structured, follows the PRISMA guidelines, and uses standard meta-analysis techniques. However, there are areas that require clarification and further major revision before it get accepted for publication.

- The abstract should mention the increased adverse reactions observed to avoid overstating the potential benefits of theophylline.

- The manuscript addresses an important question regarding the role of theophylline as adjunct therapy in COPD, but the introduction should provide more justification for this research, considering known risks of theophylline.

- Significant heterogeneity was observed in key outcomes (e.g., symptom scores, adverse reactions). The manuscript should address potential sources and perform subgroup analyses to explore this further.

- The manuscript needs to better discuss the clinical risks posed by the high rate of adverse reactions, particularly in the context of theophylline’s known narrow therapeutic window.

- The non-significant results for FEV1/FVC% and symptom scores should be more clearly interpreted to guide clinicians on their clinical relevance.

- More details on how comorbidities were handled in the included studies are necessary for transparency in the methodology.

- The lack of allocation concealment in some studies should be thoroughly discussed as a limitation that could affect the robustness of the findings.

- The discussion should be expanded to clarify where, if at all, theophylline should fit in current COPD treatment guidelines, especially given the availability of more effective therapies.

- The conclusion should highlight specific research gaps, such as the need for studies on long-term outcomes or focusing on different COPD phenotypes.

Reviewer #2: 1. What are the inclusion and exclusion criteria for patients with COPD with/without other complications? Needs to be well defined

2. Table 2 needs to be re-looked into. There are mismatches between cardiovascular and neurological adverse reactions. Besides these common symptoms, adverse reactions, and drug-drug interactions may be included.

3. Many times, COPD patients are treated with other drugs, including antibiotics. Such complications can be discussed in the discussion.

6. PLOS authors have the option to publish the peer review history of their article (what does this mean? ). If published, this will include your full peer review and any attached files.

**Do you want your identity to be public for this peer review?** For information about this choice, including consent withdrawal, please see our Privacy Policy .

Reviewer #1: No

Reviewer #2: No

---

## [Author Response · Author response to Decision Letter 1]

7 Jan 2025

Reviewer #1: The manuscript by Yang et al presented a systematic review and meta-analysis on the effects of additional oral theophylline in conjunction with inhaled therapy in stable COPD patients. The topic is an interesting one in clinical relevance as theophylline is applied in COPD for very long time. The study is well-structured, follows the PRISMA guidelines, and uses standard meta-analysis techniques. However, there are areas that require clarification and further major revision before it get accepted for publication.

(1) The abstract should mention the increased adverse reactions observed to avoid overstating the potential benefits of theophylline.

Respond: thank you for reviewer's comments. We have added the mention in abstract.

(2) The manuscript addresses an important question regarding the role of theophylline as adjunct therapy in COPD, but the introduction should provide more justification for this research, considering known risks of theophylline.

Respond: thank you for reviewer's comments. Indeed, emphasizing the risks of theophylline is very important and could further illustrate the importance of our research. So, we have added relevant content in the introduction.

(3) Significant heterogeneity was observed in key outcomes (e.g., symptom scores, adverse reactions). The manuscript should address potential sources and perform subgroup analyses to explore this further.

Respond: thank you for reviewer's comments. I am so sorry for my mistake that I mistakenly labeled the meta-analysis figure of adverse reactions. I have corrected it in manuscript and there was no Significant heterogeneity in adverse reactions. Indeed, there was Significant heterogeneity in symptom scores. The symptom scores were subjectively evaluated and recorded by researchers and there were differences between different studies that may lead to significant heterogeneity. It is difficult to perform subgroup analyses because of that It is difficult to classify the included literature.

(4) The manuscript needs to better discuss the clinical risks posed by the high rate of adverse reactions, particularly in the context of theophylline’s known narrow therapeutic window.

Respond: thank you for reviewer's comments. We have added the discussion in manuscript about the clinical risks posed by the high rate of adverse reactions as following: The therapeutic window of theophylline is relatively narrow (520 g/ml) and show high rate of adverse reactions even though within the therapeutic window (1520 g/ml) such as nausea、vomiting、irritability and insomnia. When the serum concentration exceeds 20g/ml, tachycardia and arrhythmia may occur. Fever、dehydration、convulsions and even death caused by respiratory and cardiac arrest may occur when serum concentration exceeds 40g/ml. In addition, many factors could affect the serum concentration of theophylline, such as age、medication and disease. Therefore, clinician need to be in cautions when prescribing theophylline and it is recommended to monitor the serum concentration.

(5) The non-significant results for FEV1/FVC% and symptom scores should be more clearly interpreted to guide clinicians on their clinical relevance.

Respond: Thank you for reviewer's comments. Our meta-analysis indicated non-significant results for FEV1% pred、FEV1/FVC% and symptom scores. FEV1% pred and FEV1/FVC% were important index to evaluate lung function and the effectiveness of medication therapy. Theophylline is a weak acting bronchodilator and adding theophylline on the basis of potent inhaled bronchodilators may do not produce additional effects, inducing no improvement in FEV1% pred and FEV1/FVC%. Symptoms scores are closely related to lung function and non-significant FEV1% pred and FEV1/FVC% improvements result non-significant symptom scores. The above content was added into the discussion of the manuscript.

(6) More details on how comorbidities were handled in the included studies are necessary for transparency in the methodology.

Respond: Thank you for reviewer's comments. Indeed, comorbidities may affect the effectiveness of COPD treatment. The included studies excluded patients with comorbidities that could affect the evaluation of treatment efficacy, such as bronchiectasis、tuberculosis、cancer、heart failure and infections. For some comorbidities, such as hypertension、osteoporosis, it is expected that they will not obviously affect the evaluation of treatment efficacy and have not been excluded, and there is no baseline characteristic difference between the treatment group and the control group.

(7)The lack of allocation concealment in some studies should be thoroughly discussed as a limitation that could affect the robustness of the findings.

Respond: Thank you for reviewer's comments. Allocation concealment is an important step in randomized controlled trials, aimed at ensuring the confidentiality of the randomization process and preventing researchers from predicting the grouping of subjects before allocation, thereby reducing selection bias. If no allocation concealment scheme, researchers may make subjective judgments during randomization, leading to certain subjects being preferentially assigned to specific groups and resulting in selection bias. The results of meta-analysis depend on the quality of the included studies and the reliability of the randomization process. The lack of allocation concealment schemes in the original research would affect the objectivity and accuracy of the meta-analysis. The above content was added into the discussion of the manuscript.

(8)The discussion should be expanded to clarify where, if at all, theophylline should fit in current COPD treatment guidelines, especially given the availability of more effective therapies.

Respond: Thank you for reviewer's comments. Our Meta analysis results show that adding theophylline to inhalation therapy can reduce the risk of acute exacerbation in patients with COPD. Therefore, the theophylline may be more suitable for patients who have a high risk of acute exacerbation especially when have already been treated with ICS+LABA+LAMA.

(9)The conclusion should highlight specific research gaps, such as the need for studies on long-term outcomes or focusing on different COPD phenotypes.

Respond: Thank you for reviewer's comments. We have added relevant contents in the conclusion of the manuscript.

Reviewer #2:

1.What are the inclusion and exclusion criteria for patients with COPD with/without other complications? Needs to be well defined.

Respond: Thank you for reviewer's comments. The inclusion criteria included:(1) stable COPD; (2) age>18 years old; (3) RCTs; (4) no limitations in gender, age, ethnicity, or treatment duration. (5) without bronchiectasis、tuberculosis、cancer、heart failure and infections. Exclusion criteria included: (1) non-stable COPD; (2) non-RCTs; (3) Valid ending data unable to be extracted; (4) Full text of the study is not available. (5) with bronchiectasis、tuberculosis、cancer、heart failure and infections. We have added in the manuscript.

2.Table 2 needs to be re-looked into. There are mismatches between cardiovascular and neurological adverse reactions. Besides these common symptoms, adverse reactions, and drug-drug interactions may be included.

Respond: Thank you for reviewer's comments. I am so sorry for my mistake of mismatching between cardiovascular and neurological adverse reactions. We have corrected it in the manuscript. Theophylline is metabolized by cytochrome P450 mixed function oxidases. Many other drugs can modify theophylline. For example, macrolides and quinolones could decline the clearance of theophylline and may lead to an increase of serum concentration, rising the risk of adverse reactions of theophylline. The content above was added in the discussion of the manuscript.

3.Many times, COPD patients are treated with other drugs, including antibiotics. Such complications can be discussed in the discussion.

Respond: Thank you for reviewer's comments. Stable COPD generally does not require antibiotics. When COPD patients experience acute exacerbation, short-term use of antibiotics is possible and is not expected to have a significant impact on research.

---

## [Decision Letter · Decision Letter 1]

6 Feb 2025

PONE-D-24-33914R1Effects of additional oral theophylline with inhaled therapy in patients with stable chronic obstructive pulmonary disease: A systematic review and meta-analysisPLOS ONE

Dear Dr. Yang,

Thank you for submitting your manuscript to PLOS ONE. After careful consideration, we feel that it has merit but does not fully meet PLOS ONE’s publication criteria as it currently stands. Therefore, we invite you to submit a revised version of the manuscript that addresses the points raised during the review process.

We look forward to receiving your revised manuscript.

Kind regards,

Shailza Singh, Ph.D

Academic Editor

PLOS ONE

Journal Requirements:

Additional Editor Comments:

Please revise as per reviewer’s suggestions

Reviewers' comments:

Reviewer's Responses to Questions

**Comments to the Author**

1. If the authors have adequately addressed your comments raised in a previous round of review and you feel that this manuscript is now acceptable for publication, you may indicate that here to bypass the “Comments to the Author” section, enter your conflict of interest statement in the “Confidential to Editor” section, and submit your "Accept" recommendation.

Reviewer #2: All comments have been addressed

Reviewer #3: All comments have been addressed

2. Is the manuscript technically sound, and do the data support the conclusions?

Reviewer #2: (No Response)

Reviewer #3: Yes

3. Has the statistical analysis been performed appropriately and rigorously?

Reviewer #2: (No Response)

Reviewer #3: No

4. Have the authors made all data underlying the findings in their manuscript fully available?

Reviewer #2: (No Response)

Reviewer #3: Yes

5. Is the manuscript presented in an intelligible fashion and written in standard English?

Reviewer #2: (No Response)

Reviewer #3: Yes

6. Review Comments to the Author

Reviewer #2: (No Response)

Reviewer #3: The authors have addressed most of the comments and significant improvement in the manuscript. However, I request the authors further clarify and refine before its get accepted for publication.

− The author have acknowledged heterogeneity in symptom scores, the justification for not performing subgroup analyses could be expanded. If subgroup analyses are truly infeasible, please consider alternative approaches, such as sensitivity analyses or meta-regression, to explore the sources of heterogeneity.

− It has been mentioned difficulties in classifying literature for subgroup analysis. It would be helpful to elaborate on these difficulties in the methods or discussion section. For example, were there inconsistencies in study design, population characteristics, or outcome measures that prevented meaningful subgroup comparisons?

− The authors have improved the discussion regarding the non-significant changes in lung function parameters (FEV1/FVC%), further elaboration on the clinical implications would be beneficial. How should clinicians interpret these findings in the context of COPD management, particularly in patients who are already receiving triple therapy?

− The authors have acknowledged the lack of allocation concealment in some studies as a potential limitation, which is appreciated. However, a brief discussion on how this might have influenced your results (e.g., overestimation of theophylline’s benefits) would further enhance transparency.

− The conclusion has added the research gaps. However, adding more specific recommendations (e.g., the need for long-term studies assessing cardiovascular risks or theophylline's role in different COPD phenotypes) would make the conclusion and manuscript for further improvement.

7. PLOS authors have the option to publish the peer review history of their article (what does this mean? ). If published, this will include your full peer review and any attached files.

**Do you want your identity to be public for this peer review?** For information about this choice, including consent withdrawal, please see our Privacy Policy .

Reviewer #2: No

Reviewer #3: No

---

## [Author Response · Author response to Decision Letter 2]

10 Mar 2025

Reviewer#3: The authors have addressed most of the comments and significant improvement in the manuscript. However, I request the authors further clarify and refine before its get accepted for publication.

(1) The author has acknowledged heterogeneity in symptom scores, the justification for not performing subgroup analyses could be expanded. If subgroup analyses are truly infeasible, please consider alternative approaches, such as sensitivity analyses or meta-regression, to explore the sources of heterogeneity.

Respond: thank you for reviewer's comments. When conducting a meta-analysis of symptom scores, only three studies were included, making subgroup analysis difficult. Sensitivity analysis using Stata14 software revealed that after excluding the Subramanian2015 study, the heterogeneity was significantly reduced, yet there remained no significant difference in symptom scores between the two groups (MD = -0.14, 95% CI: -0.44 to 0.15, p = 0.35, I² = 37%). The study of Subramanian2015 specifically reported night symptom scores, which may explain the heterogeneity compared to the other two studies. We have added the content in the discussion section in manuscript.

(2)It has been mentioned difficulties in classifying literature for subgroup analysis. It would be helpful to elaborate on these difficulties in the methods or discussion section. For example, were there inconsistencies in study design, population characteristics, or outcome measures that prevented meaningful subgroup comparisons.

Respond: thank you for reviewer's comments. Indeed, meaningful subgroup analyses were not conducted due to challenges in classifying the literature with regard to inconsistencies in study design, population characteristics, and medication regimens.We have added the content in the discussion section in manuscript.

(3)The authors have improved the discussion regarding the non-significant changes in lung function parameters (FEV1/FVC%), further elaboration on the clinical implications would be beneficial. How should clinicians interpret these findings in the context of COPD management, particularly in patients who are already receiving triple therapy?

Respond: thank you for reviewer's comments. FEV1/FVC%<70% serves as a key diagnostic criterion for COPD. A persistent decline in FEV1/FVC% during follow-up monitoring of COPD also indicates progressive deterioration of lung function. Meta-analysis findings demonstrate that the addition of theophylline to inhaled therapy do not significantly improve FEV1/FVC%. Although theophylline exhibits bronchodilatory effects, its efficacy is relatively modest. When combined with novel potent bronchodilators (e.g., LABA and LAMA) in COPD treatment regimens, theophylline may not exhibit additional bronchodilation and therefore fails to improve pulmonary function parameters. However, meta-analyses have demonstrated that adjunctive theophylline therapy may reduce acute exacerbation risks, primarily attributable to its anti-inflammatory properties rather than bronchodilatory mechanisms. Therefore, theophylline is recommended for patients with high risk of acute exacerbations, particularly for those already on triple therapy, but is not recommended for pulmonary function enhancement in COPD management.

(4)The authors have acknowledged the lack of allocation concealment in some studies as a potential limitation, which is appreciated. However, a brief discussion on how this might have influenced your results (e.g., overestimation of theophylline’s benefits) would further enhance transparency.

Respond: thank you for reviewer's comments. The absence of allocation concealment in primary studies may systematically compromise the validity of meta-analysis, particularly through inflated treatment effect estimates (e.g., overestimation of theophylline’s benefits) . We have added relative content in the manuscript according to reviewer's suggestion.

(5)The conclusion has added the research gaps. However, adding more specific recommendations (e.g., the need for long-term studies assessing cardiovascular risks or theophylline's role in different COPD phenotypes) would make the conclusion and manuscript for further improvement.

Respond: thank you for reviewer's comments. We have added relative content in the manuscript according to reviewer's suggestion.

---

## [Editor Report · Decision Letter 2]

17 Mar 2025

Effects of additional oral theophylline with inhaled therapy in patients with stable chronic obstructive pulmonary disease: A systematic review and meta-analysis

PONE-D-24-33914R2

Dear Dr. Yang,

We’re pleased to inform you that your manuscript has been judged scientifically suitable for publication and will be formally accepted for publication once it meets all outstanding technical requirements.

Kind regards,

Shailza Singh, Ph.D

Academic Editor

PLOS ONE
---

## [Editor Report · Acceptance letter]

PONE-D-24-33914R2

PLOS ONE

Dear Dr. Yang,

I'm pleased to inform you that your manuscript has been deemed suitable for publication in PLOS ONE. Congratulations! Your manuscript is now being handed over to our production team.

Kind regards,

on behalf of

Dr. Shailza Singh

Academic Editor

PLOS ONE